# Seeing Beyond Redundancy: Task Complexity's Role in Vision Token Specialization in VLLMs

## Abstract

Vision capabilities in vision large language models (VLLMs) have consistently lagged behind their linguistic capabilities. In particular, numerous benchmark studies have demonstrated that VLLMs struggle when fine-grained visual information or spatial reasoning is required. However, we do not yet understand exactly why VLLMs struggle so much with these tasks relative to others. Some works have focused on visual redundancy as an explanation, where high-level visual information is uniformly spread across numerous tokens and specific, fine-grained visual information is discarded. In this work, we investigate this premise in greater detail, seeking to better understand exactly how various types of visual information are processed by the model and what types of visual information are discarded. To do so, we introduce a simple synthetic benchmark dataset that is specifically constructed to probe various visual features, along with a set of metrics for measuring visual redundancy, allowing us to better understand the nuances of their relationship. Then, we explore fine-tuning VLLMs on a number of complex visual tasks to better understand how redundancy and compression change based upon the complexity of the data that a model is trained on. We find that there is a connection between task complexity and visual compression, implying that having a sufficient ratio of high complexity visual data is crucial for altering the way that VLLMs distribute their visual representation and consequently improving their performance on complex visual tasks. We hope that this work will provide valuable insights for training the next generation of VLLMs.

## 1 Introduction

Large language models (LLMs) have revolutionized the field of natural language processing, obtaining impressive performance across many complex tasks. More recently, many state-of-the-art LLMs, such as OpenAI's GPT (OpenAI, 2024), Anthropic's Claude (Anthropic, 2025), and Meta's Llama (Grattafiori et al., 2024) have been expanded to include vision capabilities as well, where vision and text tokens are embedded alongside each other and passed through the model simultaneously. One would hope that the vast knowledge that these models acquire from the billions of text tokens that they are trained on would transfer to the vision domain, resulting in similarly strong performance across many computer vision tasks. However, this has often not borne out in practice. Indeed, the vision capabilities of these models have always seemed to lag behind their linguistic counterparts (Kamath et al., 2023; Rahmanzadehgervi et al., 2024).

More specifically, prior work has identified fine-grained object recognition and spatial reasoning as two visual capacities that are especially limited in modern VLLMs (Kamath et al., 2023; Deitke et al., 2025), where a model might be able to achieve near state-of-the-art performance on image captioning or visual question answering (VQA) (Alayrac et al., 2022; Yuan et al., 2021; Li et al., 2023), only to barely perform better than chance on benchmarks that have been specifically constructed to target these deficiencies. While benchmarking is effective in demonstrating these deficiencies, it only offers high level descriptive insights into the types of problems that VLLMs struggle with.

Recent work has taken a model-focused approach to diagnosing these issues through ablation studies, visualization tools, and more. However, these works tend to highlight issues, such as token

redundancy, holistically rather than directly tying them to specific tasks. In this work, we seek to bridge this gap, explicitly focusing on fine-grained object grounding and spatial reasoning, and how a model's visual and multimodal representations directly contribute to degraded performance. Understanding this relationship will yield valuable metrics and insights for VLLM training, enabling practitioners to select architectures and datasets that lend themselves to improved performance.

We begin by introducing metrics that we use to compute the visual redundancy, or compression, of a VLLM. These include token-based metrics, such as the stable rank of the visual embeddings at the final layer, ablation-based metrics, where a percentage of visual tokens are removed and performance impacts are observed, and token probes, where a linear probe is trained on individual token positions to predict various visual features contained in the image. We run these metrics across a variety of datasets that explicitly include a wide range of image complexity, including a synthetic dataset that we construct for this purpose, and subsets of MSCOCO (Lin et al., 2014) and Whats Up? (Kamath et al., 2023). We demonstrate across these datasets, using more recent VLLMs, that *VLLMs distribute visual information across the embeddings, resulting in a high degree of uniformity across the tokens*, where each represents all objects in the image to a certain extent, limiting the existence of specialized tokens.

To demonstrate why this is problematic and directly relate it to image/task complexity, we then correlate each of the redundancy metrics that we computed to various measures of task complexity, such as the number of objects, number of unique shapes and colors, etc. We show that *high visual redundancy and limited compressability is strongly associated with reduced performance on complex tasks*, such as object counting. Unfortunately, because redundancy is ubiquitously prevalent VLLMs, this simply results in poor performance on complex tasks.

In the last section of our work, we seek to understand the role that fine-tuning plays in visual redundancy. We explore fine-tuning a VLLM on each of the aforementioned datasets, specifically analyzing the difference of outcomes between datasets which require spatial reasoning and those that require grounding. We then re-run our suite of metrics to see how redundancy and token utilization changes depending on the complexity of the fine-tuning task. We find that the type of complexity (grounding vs. spatial reasoning) changes how the model's hidden representations are altered, with the former producing larger changes in hidden states. Additionally, we find that fine-tuning a VLLM overwhelmingly alters the text representations of the model relative to the vision representations, which are largely left unaltered. These results suggest that including complex visual examples in the training data is crucial for altering the behavior of VLLMs and that the type of task complexity that is introduced can impact the hidden representations of the model at differing points.

A summary of our novel contributions are as follows:

- We provide a thorough analysis of visual redundancy in VLLMs that exceeds prior works both in breadth and depth.
- We explicitly link visual redundancy in these models to quantitative measures of task complexity, revealing detailed insights into failure scenarios.
- We investigate the impact of fine-tuning on visual redundancy, providing strong concrete takeaways that can be used in training or compressing VLLMs.

## 2 BACKGROUND AND RELATED WORK

**Vision Large Language Models**   The first vision language models to gain substantial attention and wide-spread usage were constrastively trained (Radford et al., 2021). While these models are powerful, learning robust multimodal embeddings that can be utilized across multiple tasks, one of their primary drawbacks is that they do not allow for open-ended natural language generation. More recently, vision components have been integrated into large language models (LLMs) and trained autoregressively on next-token prediction tasks, which we will refer to as vision large language models (VLLMs), or are otherwise known as large vision-language models (LVLMs). Some examples of models that follow this paradigm include Llama 3.2, LlaVa, Molmo, and Flamingo (Grattafiori et al., 2024; Liu et al., 2023; Deitke et al., 2025; Alayrac et al., 2022). VLLMs allow for open-ended natural language generation and therefore can be applied across a wide-range of tasks in zero-shot settings that constrastively trained models cannot. However, the performance of VLLMs across visual tasks is underwhelming relative to textual tasks, especially when complex reasoning is

required. Furthermore, while contrastive models are explicitly trained to align modalities, VLLMs are not, making vision-text alignment and multimodal interpretability difficult, and motivating the investigations in our work.

Let us define a generic VLLM more formally as follows. Consider a sequence of input text tokens $x^{(t)} = \{x_1^{(t)}, x_2^{(t)}, \ldots, x_{N_t}^{(t)}\}$ and an image input $x^{(i)} \in \mathbb{R}^{H \times W \times C}$. Both inputs undergo separate embedding procedures. Commonly, this will consist of pre-trained embedding layers or models for each respective modality, resulting in a final sequence of text embeddings $\mathbf{E}^{(t)} \in \mathbb{R}^{N_t \times d}$ and image embeddings $\mathbf{E}^{(i)} \in \mathbb{R}^{N_i \times d}$. These embeddings are then processed using a transformer-based decoder. In many cases, the embeddings are simply concatenated with one another and passed jointly through the model, or the text embeddings are passed to the decoder and cross-attention is utilized to integrate visual information throughout the model.

**Visual-token Compression and Redundancy**  It is well established that not all of the vision tokens are needed in VLLMs to obtain strong performance across various tasks. For instance, Kaduri et al. (2025) found that randomly ablating up to 95% of the tokens from LlaVA resulted in a minimal drop in performance on COCO. However, much of the effort in this space has focused on exploiting this redundancy rather than understanding it. This has led to works such as Ye et al. (2025); Gholami et al. (2025); Dhouib et al. (2025); Yang et al. (2025); Xing et al. (2024), which propose mechanisms for selecting which visual tokens to discard, further restricting the number of tokens while retaining performance. More recently, a few other works have sought to understand how visual information propagates through the model (Zhang et al., 2025; Yin et al., 2025). In this work, we build on these studies, extending them in a number of novel directions. Firstly, we are the first work to explore many of the more recent, high performing VLLMs, such as Molmo and Llama 3.2, whereas most prior works have focused exclusively on LlaVA. Furthermore, we are the first work to focus on evaluating multiple models across the same tasks, using the same metrics, simultaneously. Additionally, we are the first work to explore additional fine-tuning and the impact that this has on visual redundancy/compression, offering insights into how data affects these aspects of a model.

## 3 METHODS

### 3.1 COMPRESSION METRICS

We consider two types of compression metrics, those focused on token norms and those focused on token rank. We explore 3 different metrics for each type. For token norm metrics, we utilize the Gini coefficient, which quantifies the inequality of $L_2$ norms across the token embeddings, the normalized entropy, which quantifies the information content of the token norm distribution, and the coefficient of variation (CV), which measures the relative variability of token norms. While the norm metrics provide interesting insights regarding how information is distributed across tokens, it does not provide a means of further characterizing exactly how much information is stored across the tokens. To this end, we also utilize rank metrics, which are computed over the token matrix of a given layer. The first metric that we use is the stable rank, which measures the effective dimensionality of the token matrix, the participation ratio, which measures how many singular values contribute significantly to the collective representation of the tokens at a given layer, and the exponential entropy, which computes the Shannon entropy of the singular value distribution. More details for each metric are available in the Appendix.

### 3.2 SVD ANALYSIS

To more deeply explore the relationship between vision, text, and multi-modal hidden states, and the impact their content and interaction has on final performance, we choose to apply singular-value decomposition analysis to both the unimodal and multimodal hidden state matrices. To do so, we first extract the text and vision tokens from the multimodal matrix. We then apply SVD to the vision $(U_v, S_v, V_v^T)$, text $(U_t, S_t, V_t^T)$, and multimodal matrices $(U_c, S_c, V_c^T)$ individually, producing the building blocks for our analyses.

**Alignment**  We first look at *token projection consistency*, a measure of how consistently the primary unimodal token component projects onto the primary token component of each modality in

multimodal token space. To compute this, we take the dot product between the normalized primary token component of each modality in multimodal space and the primary token component of each unimodal matrix. Such a metric allows us to understand whether unimodal tokens are able to collaborate in multimodal space (high consistency), or if individual modes retain specialized communication with each other (low consistency). We next look at *feature space alignment*, or the dot product between the top right singular vectors of unimodal and multimodal matrices. This metric provides us a sense of whether primary unimodal token representations are able to fuse in the multimodal setting. Next we look at *subspace alignment*, or the degree to which each modality contributes to the primary token component of the multimodal matrix. For a given modality $m$, we simply take the average per-token energy of each modality's primary token component $e_m = \frac{\|u_{c_m}\|^2}{n_m}$ and then normalize them by the average energies to obtain alignment: $a_m = \frac{m_{pt}}{m_{pt} + \neg m_{pt}}$. This metric helps us understand whether vision or text tokens dominate the global token pattern when modal counts are accounted for. Lastly, we look at *reconstruction alignment*, or the ability of primary unimodal feature-space representations to reconstruct the multimodal feature-space representations. To compute alignment, we simply use the projection matrix formed by the first $k$ right singular vectors of each modal SVD, $V_k V_K^T$ to reconstruct the original multimodal token matrix $Q$ via $Q V_K V_K^T$. For a given modality, we then compute the coefficient of determination between the original matrix and its reconstruction via:

$$R^2 = 1 - \frac{mean(Q - Q V_K V_K^T)^2}{var(Q)}.$$

For our reconstructions, we compare results using an arbitrary rank ($k = 5$). See the Appendix for more details on our SVD analyses.

### 3.3 LINEAR PROBING

While the compression metrics offer a high-level overview of token compression at each layer of the model, they do not provide insights into the specific information that is stored in individual tokens. To explore this aspect of the models' embeddings, we employ MLP probes, trained on the embeddings to predict various visual features of the image, e.g., the most common object in the image or the largest object in the image. To ensure that the number of tokens is consistent across individual datasets, we resize all images to a uniform size. Then, we train a probe for each individual position across each layer of the model. Each probe is a MLP with two hidden layers and ReLU activations. The specific visual features that each probe predicts varies by dataset, but all features are framed as classification problems. By looking at the mean accuracy and variance in accuracy across layers, these probes allow us to capture general trends about which layers encoded which types of information, as well as whether tokens are specialized to store specific types of visual information vs. generally storing global image information.

### 3.4 TOKEN ABLATIONS

The final metrics we use in our analysis are ablation-based. Here, we directly assess the performance of the model on the downstream task as tokens are randomly ablated from the model. This most directly corresponds to how VLLMs are compressed and offers a global perspective on how each individual token that enters the model is impacting overall performance. While complex token ablation methodologies have been proposed (Ye et al., 2025; Gholami et al., 2025; Dhouib et al., 2025), for consistency across models and datasets, we adopt a simple methodology. We define an ablation ratio $\rho$, which corresponds to the percentage of tokens dropped from the model's inputs. For a given ablation ratio, we randomly select $N_\rho$ tokens to drop from the visual tokens, where $N_\rho = \rho * N$ and $N$ is the total number of visual tokens in the model's input.

## 4 EXPERIMENTS

### 4.1 ZERO-SHOT ANALYSES

We begin by analyzing two VLLMs in a zero-shot setting leveraging the metrics described in the previous section. We selected two recent VLLMs (at the time of beginning our experiments) Molmo

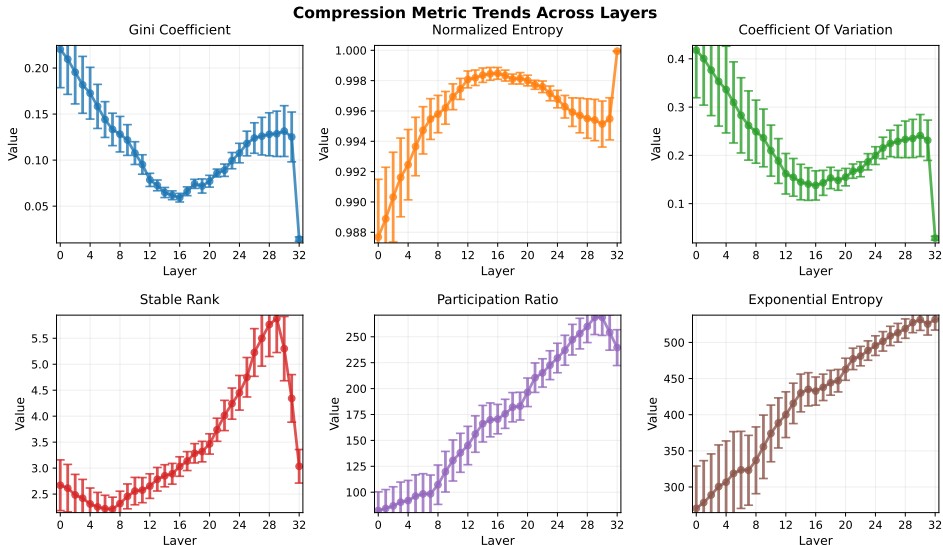

Figure 1: Visual compression trends across layers of Molmo for our proposed synthetic dataset.

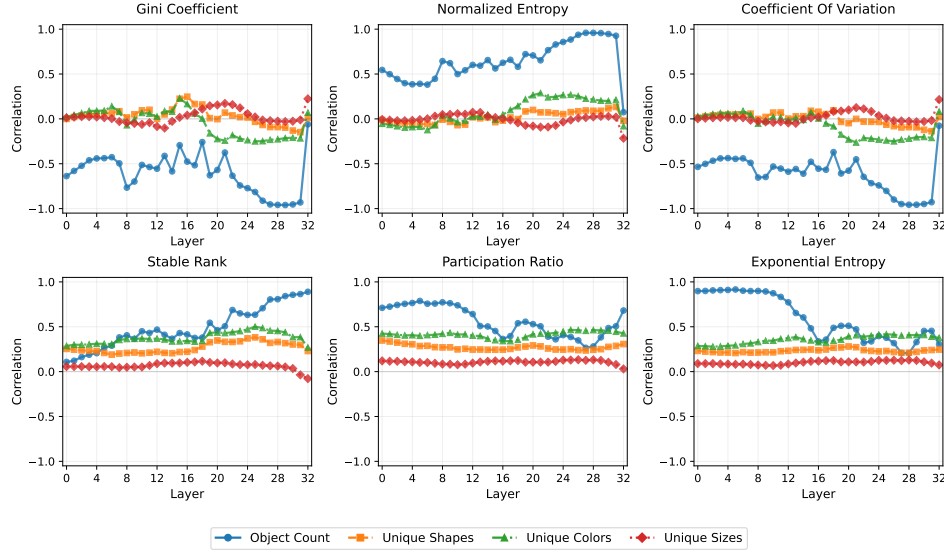

Figure 2: Spearman correlation between various compression metrics and various visual attributes across layers of Molmo for our proposed synthetic dataset.

(Deitke et al., 2025) and Llama 3.2 (Grattafiori et al., 2024). Both models achieve impressive performance across a variety of tasks and are fully open-source, making them strong candidates. Additionally, they have slightly different architectures, providing a broad coverage of most modern VLLM models. Namely, Molmo prepends the vision tokens to the text tokens, passing them through a transformer decoder jointly, allowing for multimodal attention at all layers of the model. Llama 3.2 instead leverages cross-attention to combine vision and text, where text tokens are passed through the transformer decoder and at fixed intervals, cross-modal attention is allowed.

**Datasets** When quantifying complexity in images, there are many factors that must be considered, especially in real-world complex scenes, including the number of objects, overall size of objects, size of objects relative to one another, color of objects, number of colors in each object, number of colors in the background, amount of background, etc. To avoid this complexity in our initial experiments, we opted to generate a synthetic image dataset using python, consisting of 2D shapes on a white

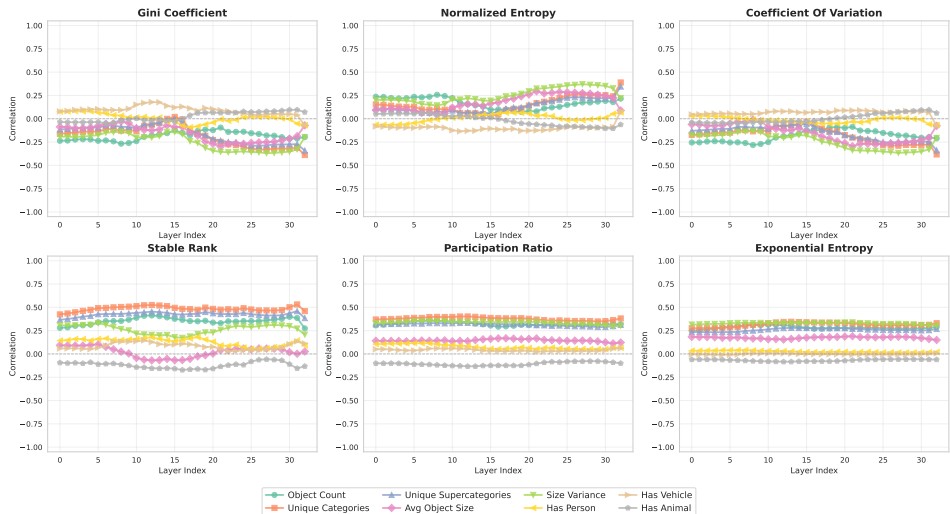

Figure 3: Spearman correlation between various compression metrics and various visual attributes across layers of Molmo for our COCO.

background. We varied complexity across a number of fixed axes, allowing for complexity to be more precisely quantified in each image. More specifically, we start with a base configuration which consists of a single object of random shape, color, and size placed on a white background, and using this base configuration we generate families of images at multiple object counts, ranging from 1-200. For each count, it produces that number of objects with only the baseline configuration, then it systematically varies one aspect at a time (shape, color, or size) as well as a mixture of each. We leverage a visual vocabulary consisting of 5 shapes, 10 colors, and 3 sizes, allowing for a variety of images to be generated for a single base configuration. For this work, we generated images of 1000x1000 with 20 base configurations, resulting in 8,220 images in total.

While the synthetic data is most conducive to our analyses due to the fact that we can carefully control for image complexity and isolate specific variables, we also want to verify that similar trends hold for real data. Therefore, we run similar experiments on a randomly sampled subset of MSCOCO (Lin et al., 2014). More specifically, we perform our compression and probing experiments on 1000 images and our ablations on 250 images. Due to certain object information not being present in COCO, such as the color of objects, we shift to a different set of image features for our evaluation. Many are self explanatory, such as the presence or absence of various objects, the primary category, and whether multiple object categories are present in the image.

**Token Compression** Figure 1 contains the layer-wise compression results for each of our proposed metrics using Molmo and our synthetic dataset.[1] The metrics that focus on token-wise sparsity (top metrics) all follow the same pattern, where the early layers spread attention and energy across a large number of vision tokens. However, as the mid-point of the model is reached, visual information is re-concentrated into a smaller number of tokens. As the end of the model is reached, information is again spread more evenly across tokens, with the final sharp change in compression being due to the hidden states at the final layer being extracted post-layer norm. The bottom three metrics, which focus on the dimensionality of the visual information show a slightly different pattern, where over the course of the model, the effective dimensionality of the visual information increases as they represent increasingly complex visual features. However, there is a sudden drop off towards the end where dimensions are collapsed for efficient read-out, potentially discarding some information that is useful to the model. Overall, this paints a picture of visual processing in which information is spread across tokens at the beginning of the model, and the tokens are representing relatively simple visual concepts. At the mid-point of the model, visual information is compressed, spreading across fewer tokens, but the effective dimensionality of the data is increasing as these tokens represent

---

[1]We do not include results for Llama 3.2 in this set of experiments because the vision tokens are fixed throughout the model.

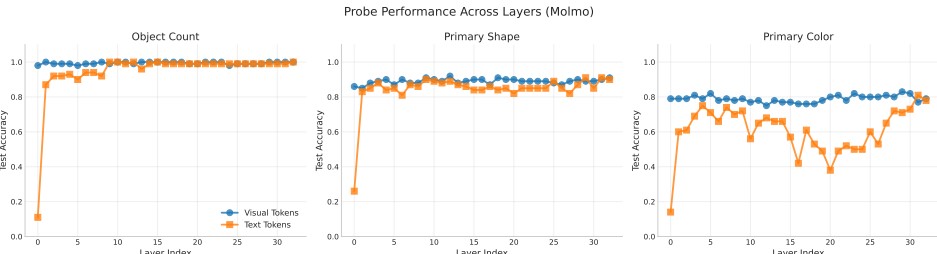

Figure 4: Linear probe performance on various visual attributes using Molmo on our synthetic dataset.

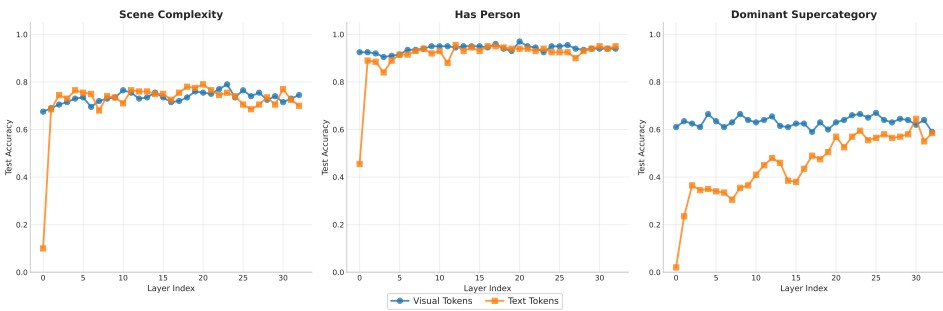

Figure 5: Linear probe performance on various visual attributes using Molmo on COCO.

increasingly complex visual concepts. Then, at the end, information is again spread across tokens and extraneous visual information is discarded. Figure 2 provides further insights into how visual compression is correlated with various attributes of the synthetic data, showing the Spearman correlation between each of the compression metrics and various visual attributes of the corresponding images. Object count is overwhelmingly correlated with less compression, suggesting that cluttered scenes need more tokens to be effectively represented. The number of unique shapes and colors in the image are mildly correlated, suggesting that some amount of object variety also requires more dimensions for effective representation. However, the number of unique object sizes does not matter for the overall compressability of the image.

Figure 3 contains the compression results for MSCOCO. Overall, the trends across layers are less noticeable and the magnitude of correlation tends to be lower. The number of unique categories and size variance stand out as the most prominent features in terms of reducing visual compression. None of the features that tested for whether specific types of categories, e.g. "has person", displayed a significant correlation with visual complexity. There are some stand-outs, such as "has animal", which had a negative correlation across most metrics, but this is likely due to the nature of the scenes which contain these types of objects. Aside from these categories, average object size stands out as another feature that has a slightly negative correlation in some of the metrics. This is likely due to the fact that when the average object size is high, there are a select few objects dominating the scene relative to others, allowing for more visual compression. However, this may also be suboptimal in some cases because if the object still has a sufficient amount of detail and the prompt is attempting to elicit specific information about the object, this could result in failure due to over compression.

**Probes**   Figure 4 contains the results for 3 different probing experiments for Molmo on our synthetic data.[2] These experiments provide further insights into how visual information is processed in the model. One notable finding is the text tokens are highly predictive of visual attributes, indicating that a substantial amount of visual information is stored in these tokens. Indeed, this translation of visual information into the text space is immediate, where after the first layer, the text tokens already achieve strong performance on each of the 3 tasks. The strongest performance is observed on object count, which aligns with the insights from the compression metric correlations, where at all layers of

---

[2]A figure for Llama 3.2 is available in the Appendix.

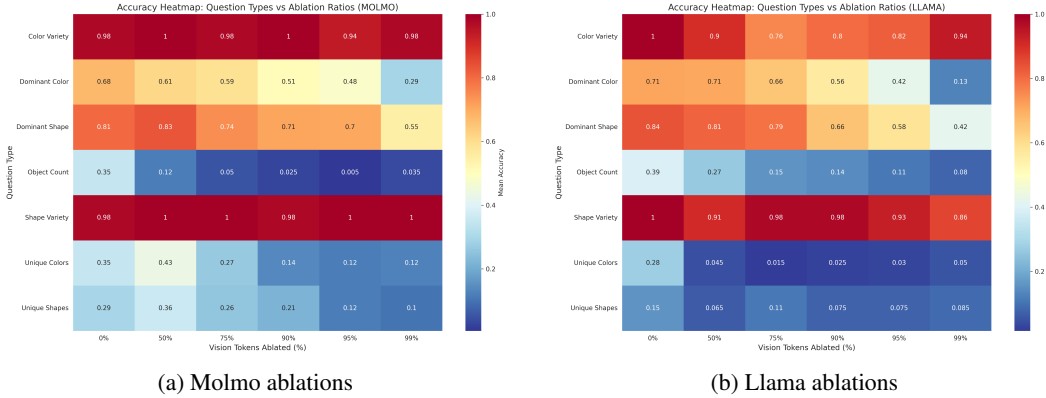

(a) Molmo ablations         (b) Llama ablations

Figure 6: Visual ablation performance across various question types using our synthetic dataset. Left: Molmo results. Right: Llama 3.2 results.

the model can predict the number of objects in the scene with high accuracy. Furthermore, because the results are averaged over all positions in the model we can see that virtually all token positions are predictive of object count and therefore contains some amount of object-level information. This points out a potential inefficiency in how visual information is processed in a VLLM and suggests that there is a high amount of redundancy present in the model. The other visual attributes are not quite as predictive as object count, suggesting that certain types of object-level information is stored across various tokens. However, again there is very little change in probe performance across layers, especially for the visual tokens. The text tokens for primary color do show some level of variance across layers, suggesting that there may be more visual specialization in the text tokens at the middle layers compared to the visual tokens at other layers.

Figure 5 contains the probe results for Molmo on COCO. Here, scene complexity is a measure with 3 classes corresponding to $\leq 2$, 3-5, and $> 5$, respectively. Similar to the synthetic data, we observe a general trend that both textual and visual features are highly predictive of visual attributes. Immediately after the first layer visual information is integrated into the textual features. However, "dominant supercategory" is a notable outlier, where the textual tokens gradually include more visual information. In the synthetic experiments, "primary color" displayed a trend that, while different, is also distinct from most of the other categories. These results suggests that not all visual information is immediately fused into the textual features, and that the model at various layers focuses more on certain visual features within the multimodal feature space.

**Visual Ablations** Figure 6a contains the visual ablation performance for both Molmo and for Llama 3.2 on our synthetic dataset. Overall, as tasks become more complex, or fine-grained, compression has a larger impact on model performance. Color and shape variety are the highest level tasks, where the model must recall which colors and shapes are present in the image. This is simple image-level information that is easily recalled, even when 99% of the visual tokens are discarded. The tokens come from CLIP before being ablated and passed into the model, meaning that information is already spread across tokens, making this observation possible. Next we have tasks such as finding the dominant color and shape, as well as the unique shapes and colors. These tasks are not as high level as the previous tasks because some object-level comparison is needed. Nevertheless, relative performance can be retained up to relatively high ablation ratios where even at 90% decent predictive performance is sustained. Lastly, the most complex task, object count, requires counting up to 200 objects in the scene. Performance on this task deteriorates quickly, halving with just 50% of the tokens discarded and reaching close to zero at 75%. At first glance, this appears to contradict the probe results, where most tokens were predictive of object count. However, this likely means that while all tokens have the information needed to answer this question, the model learns to only attend to certain tokens to extract this information. So, while the tokens themselves are not heavily specialized, the attention weights of the model are specialized. Overall, these results suggest that compression should not be uniformly applied for each task; its effectiveness is directly correlated with task complexity, where complex tasks should have a less compression than simpler tasks. A discussion of visual ablations on COCO is available in the Appendix.

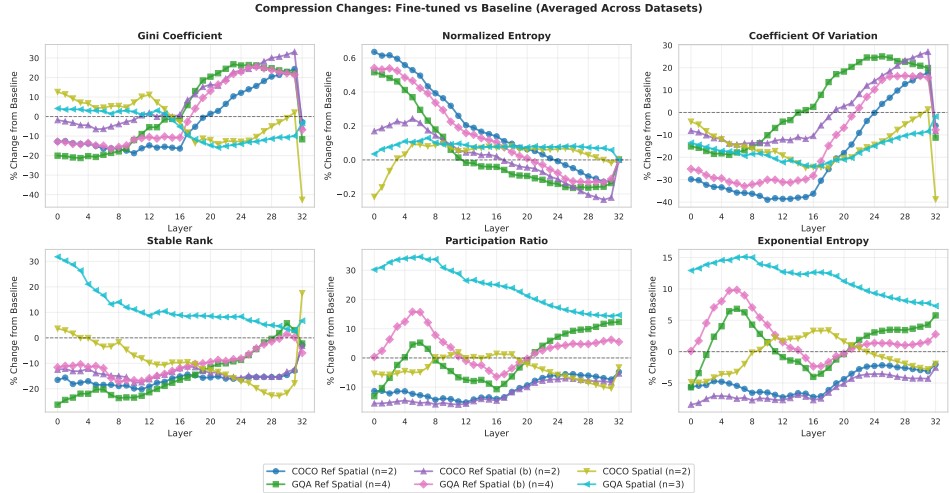

Figure 7: Visual compression trends across layers of Molmo averaged across various fine-tuning and evaluation datasets.

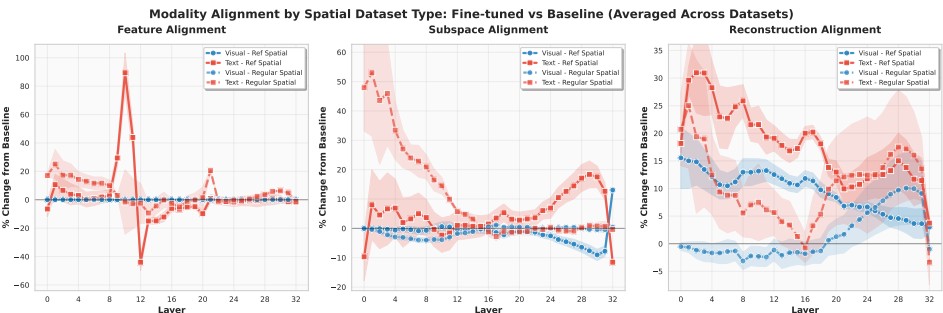

Figure 8: SVD alignment metrics for Molmo averaged over each of our fine-tuning datasets according to whether they were referring expression datasets.

## 4.2 FINE-TUNING

**Datasets** Dataset selection for our fine-tuning experiments is driven by multimodal task complexity, encouraging vision-language alignment. Therefore, we select visual referring expression and spatial reasoning as our two target tasks. We use GQA and COCO to produce 3 datasets of varying complexity for each, roughly following the procedure outlined in Kamath et al. (2023). A detailed breakdown of these datasets and the creation process is available in the Appendix.

**Visual Compression** Figure 7 contains the compression metrics averaged over each of our evaluation and fine-tuning datasets, demonstrating differences in compression throughout Molmo. Overall, depending on the nature of the fine-tuning dataset, different trends are observed. The referring expression datasets all follow a similar trend where early layers become less concentrated while also dropping in rank. At later layers, the rank stays lower than the baseline, but more specialized tokens emerge as the model begins the task of localizing the target object. In contrast, the more complex spatial reasoning datasets lead to the emergence of more specialization in early layers, with a higher stable rank, while the later layers contain less specialization. We believe this difference is due to the fact that the output is more in-line with the pre-training data distribution as it take the form of natural language for the spatial reasoning tasks, whereas the localization output, while included during pre-training, represents only a small portion of the data, requiring a larger distribution shift. These results suggest that if more specialization is desired, a combination of the two tasks is desirable, where a target must be localized while also requiring spatial reasoning at the earlier layers. This would result in increased rank and more specialized tokens throughout the model.

**SVD Alignment** Figure 8 contains the results for the SVD metrics on our fine-tuned models. We averaged over each of our datasets according to whether they were referring expression datasets or just standard spatial datasets. For feature alignment, text showed noticeable changes, while vision showed minimal change. The referring expression data produced sharper, more localized spikes in the models' layers relative to the spatial reasoning datasets. From these results, we can observe that fine-tuning mainly rotates the principal text features towards the multimodal features, leaving the vision features relatively static, and that data which is more out-of-distribution (the referring expression task is relatively distinct from a standard task due to the localization output) results in more abrupt shifts in the singular vectors. The subspace alignment again shows more substantial changes in the text representations, where text is overwhelmingly dominant in the early layers and moderately dominant in the final layers. Here the regular spatial reasoning datasets are more impactful. This is likely due to the increased complexity of the spatial reasoning prompts, which the model needs to successfully parse in order to complete the task. Lastly, reconstruction alignment again confirms that the text subspace explains substantially more of the multimodal representations across most layers of the model. Noticeably, for the referring expression data, the text explains even more of the multimodal representations and the vision explains a substantial part as well. This suggests that visual alignment with the multimodal space is more important for this task throughout the model as the model tries to localize the target object, whereas for the standard spatial reasoning dataset, visual information is not largely integrated into the multimodal space until the end, as the model needs to consider spatial information only after the prompt is sufficiently understood.

## 5 DISCUSSION

The results of our zero-shot and fine-tuning experiments reveal interesting insights into how VLLMs represent visual information and the impact that fine-tuning has on the model's internal representations. Many of our insights will prove valuable for creating better compression policies in VLLMs. Namely, we found that the most compression should occur at the earliest layers in the network, where information is most diffuse, with moderate compression at later layers, and only mild compression at middle layers where tokens display more specialization and dropping the wrong one could more substantially impact model performance. Furthermore, compression should be based on the downstream task. Simple tasks such as visual recognition can have extremely severe compression ratios. A compression method that is dynamic and can adapt to each task would naturally be most effective. In terms of effective fine-tuning, we found that updating the text and any multimodal projection layers would likely be more important for task adaptation than fine-tuning the visual encoder. This is because even in the LLM decoder, the visual features remain relatively consistent. Additionally, we found that tuning on grounding vs. spatial reasoning tasks could display different trends across the model, suggesting that mixing this data is key to performance improvement across all layers of the model. Spatial reasoning seemed to make even more substantial changes to the text representation of the model, furthering its multimodal alignment, while the grounding data seemed to do the same for vision. This validates the direction that Molmo (Deitke et al., 2025) originally pursued, which involved collecting fine-tuning data that specifically targeted both of these tasks. However, it reveals that even further expanding the data to include more of these tasks could improve multimodal representations within the model.

## 6 CONCLUSION

In this work, we presented a fine-grained analysis of VLLM redundancy and complexity. Building on prior work which primarily focused on attention distributions throughout the model, we instead focused on analyzing hidden states. Overall, we found a high-level of redundancy in the VLLMs, especially in early and late layers of the model where activations are spread across more tokens. Additionally, via a customized synthetic dataset, we discovered that complex tasks, such as object counting, require more specialized tokens to be successful and are less robust to token ablation methodologies. Lastly, we fine-tuned a VLLM on a large number of datasets and observed that different types of complexity, based on downstream task, can impact the model representations in different ways, but all data has a larger impact on text representations than vision representations. We hope that these insights will prove valuable in developing the next generation of VLLM compression methodologies and in selecting effective datasets for fine-tuning VLLMs.

## 7 ETHICS STATEMENT

All of the results in this paper are utilizing existing publicly available models and datasets. Additionally, our work is geared towards better understanding these models and impact that training on certain datasets can have on the model. Therefore, we do not foresee any notable ethical implications of our paper.

## 8 REPRODUCABILITY STATEMENT

While we do not include code with our submission due to time constraints, we do plan to release our code upon publication. Additionally, we have included detailed information about our dataset creation, metric implementation, and fine-tuning setups, including the compute that was used, in the Appendix, to encourage reproducability.

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

# A APPENDIX

## A.1 COMPRESSION METRIC DETAILS

We consider two types of compression metrics, those focused on token norms and those focused on token rank. We explore 3 different metrics for each type. For token norm metrics, we utilize the Gini coefficient, which quantifies the inequality of $L_2$ norms across the token embeddings. The coefficient ranges from 0, which indicates perfect equality across the norms and 1, which indicates maximum inequality where a single token has all of the norm mass. For a set of token norms $\{n_1, n_2, \ldots, n_k\}$, the Gini coefficient $G$ is calculated as the area between the line of equality and the Lorenz curve of the sorted norm distribution. The second token norm metric is the normalized entropy. The normalized entropy quantifies the information content of the token norm distribution, defined as $H_{\text{norm}} = -\sum_i p_i \log(p_i)/\log(k)$, where $p_i$ is the probability mass of token $i$ (computed as $n_i/\sum_j n_j$) and $k$ is the total number of tokens. Values range from 0 (deterministic distribution)

to 1 (uniform distribution), indicating the degree of concentration in the token representation space. The final norm metric is the coefficient of variation (CV), which measures the relative variability of token norms. It is computes as $\text{CV} = \sigma/\mu$ where $\sigma$ and $\mu$ are the standard deviation and mean of the $L_2$ norms respectively.

While the norm metrics provide interesting insights regarding how information is distributed across tokens, it does not provide a means of further characterizing exactly how much information is stored across the tokens. To this end, we also utilize rank metrics, which are computed over the token matrix of a given layer. The first metric that we use is the stable rank, which measures the effective dimensionality of the token matrix. The stable rank is defined as $\text{SR} = \frac{\sum_i \sigma_i^2}{\sigma_1^2}$, where $\sigma_i$ are the singular values of the token embedding matrix in descending order. This metric provides a stable estimate of the intrinsic dimensionality that is less sensitive to noise than the traditional rank, with values ranging from 1 (rank-1 matrix) to the minimum matrix dimension. The next metric is participation ratio, which measures how many singular values contribute significantly to the collective representation of the tokens at a given layer. It is computed as $\text{PR} = \frac{(\sum_i \sigma_i)^2}{\sum_i \sigma_i^2}$, where a high PR indicates that the tokens are utilizing their full dimensionality and a low PR indicates information is compressed and the tokens are utilizing fewer effective dimensions. The last norm metric is the exponential entropy, which computes the Shannon entropy of the singular value distribution, as $\text{EE} = \exp\left(-\sum_i \tilde{p}_i \log(\tilde{p}_i)\right)$ where $\tilde{p}_i = \sigma_i / \sum_j \sigma_j$.

## A.2 SVD ANALYSIS DETAILS

To more deeply explore the relationship between vision, text, and multi-modal hidden states, and the impact their content and interaction has on final performance, we choose to apply singular-value decomposition analysis to both the unimodal and multimodal hidden state matrices. To do so, we first extract the text and vision tokens from the multimodal matrix. We then apply SVD to the vision $(U_v, S_v, V_v^T)$, text $(U_t, S_t, V_t^T)$, and multimodal matrices $(U_c, S_c, V_c^T)$ individually, producing the building blocks for our analyses.

**Alignment**  We first look at *token projection consistency*, a measure of how consistently the primary unimodal token component projects onto the primary token component of each modality in multimodal token space. To compute this, we take the dot product between the normalized primary token component of each modality in multimodal space and the primary token component of each unimodal matrix. For a given modality $m$, we have

$$\text{consist}_m = \frac{u_{c_m}}{\|u_{c_m}\|} \cdot u_m$$

where $u_{i_j}$ represents the primary token component of the multimodal or unimodal matrix, the former of which is selected for a specific modality when a second subscript is present. Such a metric allows us to understand whether unimodal tokens are able to collaborate in multimodal space (high consistency), or if individual modes retain specialized communication with each other (low consistency). We next look at *feature space alignment*, or the dot product between the top right singular vectors of unimodal and multimodal matrices. For a given modality $m$, we have

$$\text{fsa}_m = \frac{v_m^T}{\|v_m^T\|} \cdot \frac{v_c^T}{\|v_c^T\|}$$

which measures the principle directions between unimodal and multimodal components in feature space. This metric provides us a sense of whether primary unimodal token representations are able to fuse in the multimodal setting. Such a metric provides additional insight into the collaboration between tokens. Next we look at *subspace alignment*, or the degree to which each modality contributes to the primary token component of the multimodal matrix. For a given modality $m$, we simply take the average per-token energy of each modality's primary token component

$$e_m = \frac{\|u_{c_m}\|^2}{n_m}$$

and then normalize them by the average energies to obtain alignment:

$$a_m = \frac{m_{pt}}{m_{pt} + \neg m_{pt}}.$$

This metric helps us understand whether vision or text tokens dominate the global token pattern when modal counts are accounted for. Lastly, we look at *reconstruction alignment*, or the ability of primary unimodal feature-space representations to reconstruct the multimodal feature-space representations. To compute alignment, we simply use the projection matrix formed by the first $k$ right singular vectors of each modal SVD, $V_k V_K^T$ to reconstruct the original multimodal token matrix $Q$ via $Q V_K V_K^T$. For a given modality, we then compute the coefficient of determination between the original matrix and its reconstruction via

$$R^2 = 1 - \frac{mean(Q - Q V_K V_K^T)^2}{var(Q)}.$$

For our reconstructions, we compare results using an arbitrary rank ($k = 5$) and the varying stable rank of each unimodal matrix. To be transparent with these reconstructions, we also highlight *concentration*, or the amount of variance captured by the top-k singular values for both unimodal and multimodal matrices via $\frac{\sum_{i=1}^{k} \sigma_i^2}{\sum_{i=1}^{r} \sigma_i^2}$.

### A.3 Prompts used for Zero-shot Experiments

The prompts used for our synthetic data zero-shot experiments are:

1. General image description prompt:
   - "Describe this image."
2. Object counting prompts:
   - "How many shapes are in this image?"
   - "How many unique shapes are there in this image?"
   - "How many unique colors are there in this image?"
3. Shape recognition prompts:
   - "What is the most common shape in this image?"
4. Color recognition prompts:
   - "What is the most common color of shape in this image?"
5. Shape variety/presence prompts:
   - "Are there both $\langle shape_1 \rangle$s and $\langle shape_2 \rangle$s in this image?"
6. Color variety/presence prompts:
   - "Can you see both $\langle color_1 \rangle$ and $\langle color_2 \rangle$ objects in this image?"

The prompts used for our COCO zero-shot experiments are:

1. General image description prompt:
   - "Describe this image."
2. Object counting prompts:
   - "How many objects are in this image?"
   - "How many different types of objects can you see in this image?"
3. Object recognition prompts:
   - "What is the main object or most prominent thing in this image?"
4. Scene classification prompts:
   - "Is this an indoor or outdoor scene?"

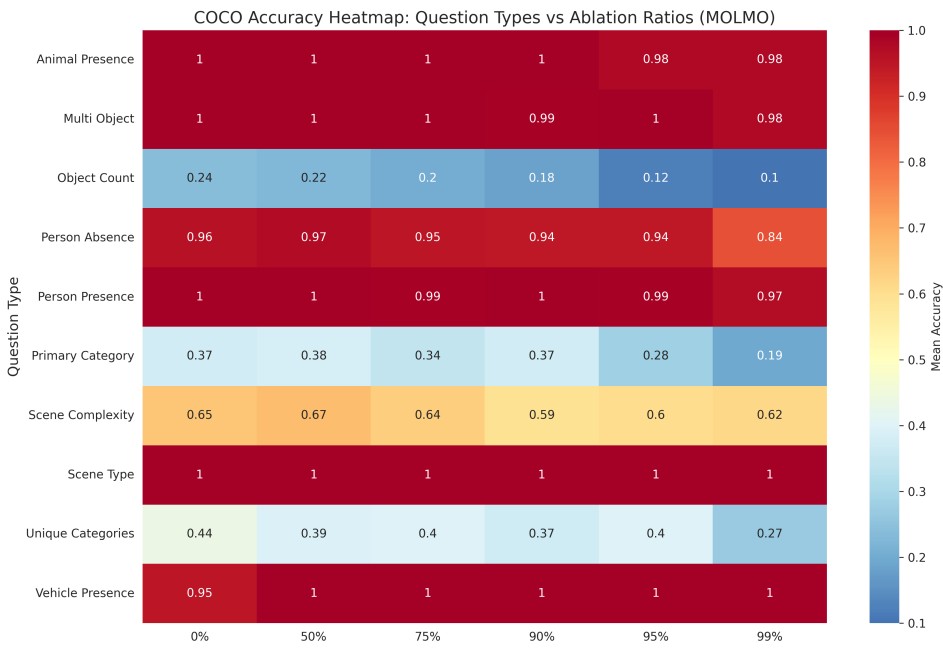

Figure 9: Visual ablation performance across various question types using Molmo on COCO.

- "Would you describe this as a simple scene with few objects or a complex scene with many objects?"

5. Object presence/absence prompts:
   - "Are there any people in this image?"
   - "Can you see any vehicles in this image?"
   - "Are there any animals in this image?"

6. Multi-object detection prompts:
   - "Can you see both $\langle category_1 \rangle$ and $\langle category_2 \rangle$ in this image?"

### A.4 COCO VISUAL ABLATIONS

Figure 9 contains the results of our visual ablation study on COCO. Similar to the synthetic data, complex tasks show a more noticeable performance degradation, highlighting the fact that general image-level information, such as the presence or absence of certain objects, is stored across numerous tokens. Notably, scene complexity is a more complex task, as it requires the model to have some understanding of the number of unique object categories and while the model struggles at this task more than the others, performance only degrades slightly. However, as tasks become increasingly complex, such as fully enumerating the number of unique categories or determining the most dominant category, requiring some additional visual reasoning, we notice a share drop in performance. A similar trend is once again observed for object counting, a task which requires fine-grained localization of objects in the image. Overall, given the number of categories that maintain their performance at high ablation levels, it is surprising how much visual information that the model compresses into relatively few tokens, highlighting the large amount of redundancy that is present in these models.

### A.5 FINE-TUNING EXPERIMENTAL DETAILS

#### A.5.1 DATASETS

Dataset selection for our fine-tuning experiments is driven by two primary interests. The first is to understand the effect which specializing a vision-language model in referring expression tasks has on the content of token representations. The second is to test the competence of vision-language

models on simple spatial reasoning tasks. For these tasks, we use GQA (Hudson & Manning, 2019) and COCO (Lin et al., 2014) to produce three vision-language tasks of varying complexity from each dataset:

1. Spatial captioning for one and two objects:
   - One-object: "Describe the spatial relationship of the $\langle obj \rangle$ relative to the image."
   - Two-object: "Describe the spatial relationship between the $\langle obj_1 \rangle$ and the $\langle obj_2 \rangle$ in the image."
2. Referring expression for one and two objects:
   - One-object: "Point to the $\langle obj \rangle$ on the bottom."
   - Two-object: "Point to the $\langle obj_1 \rangle$ to the left of $\langle obj_2 \rangle$."
3. Referring expression with localization information for two objects:
   - Two-object: "Point to the $\langle obj_1 \rangle$ to the left of $\langle obj_2 \rangle$ at $[bbox_2]$

Spatial captioning tasks vary by dataset, but ground-truths generally follow the format "A photo of a $\langle obj_1 \rangle$ to the right" for single objects and "A photo of a $\langle obj_1 \rangle$ to the right of a $\langle obj_2 \rangle$". Referring expression ground truths are simply bounding boxes of format $[x_1, y_1, x_2, y_2]$. We use bounding boxes to control for the possibility that Molmo may have seen subsets of COCO or GQA during its training. This adds a novel dimension of visual complexity, since Molmo was only trained on centerpoint referring expressions. We also mix captions into the COCO datasets, following the suggestion in Kamath et al. (2023).

For both COCO and GQA, we create spatial-captioning, referring expression, and referring expression with localization datasets. Before the creation of any dataset, we roughly follow the methods outlined in Kamath et al. (2023) by applying a broad filter to all image-question pairs. For the COCO datasets, we filter images that contain multiple annotations of the same object to avoid model confusion. We then set a box-size tolerance to exclude any objects that are too small (less than 3% of image area) and too large (greater than 30% of image area). Finally, for each image we calculate its centroid and enforce the conditions

$$|c_x^{bbox} - c_x^{im}| > \alpha \cdot w; \quad ; \quad |c_y^{bbox} - c_y^{im}| > \alpha \cdot h$$

such that no object is "centered" within the image. This is to minimize model confusion for grounding tasks since we exclusively use prepositions {left, right, above, below, bottom, top} in all referring expression tasks. For the GQA datasets, we follow a similar process. First we filter questions by the presence of target prepositions, keeping only those questions which contain our target prepositions. We then filter out all questions where mentioned objects are not present in the scene, and remove qualifiers from objects (e.g. "big", "green") to keep them aligned with the methods in Kamath et al. (2023). Finally, we remove any objects which are less than 3% of the image area. In all referring expression tasks from the second cohort, we only retain the original bounding boxes for localization, and do not create centerpoint annotations. By doing so, we require Molmo to produce completely new grounding outputs. In addition, we evaluate on six datasets produced by Kamath et al. (2023), including one-and-two object spatial variants of GQA, COCO, and their own proprietary dataset, WhatsUp. A breakdown of the dataset statistics is available in Table 1.

**Training**  Molmo 7B-O (Deitke et al., 2025) is fine-tuned separately on datasets CS, CRS, CRSB, GS, GRS, and GRSB for three epochs using cross-entropy loss. LoRA (Hu et al., 2022) is used for fine-tuning *all modules* within Molmo. We use batch-size of 2 and gradient accumulation of 2 across 8 H100 GPUs for an effective batch size of 32. We set learning rates to $lr_{\text{base}} = 5e^{-5}$, $lr_{\text{vis}} = 2e^{-6}$, and $lr_{proj} = 1e^{-5}$, and LoRA parameters to $r = 16$ and $\alpha = 32$. For referring expression datasets CRS and CRSB, regular captioning instances are included to retain language capabilities, as outlined in Kamath et al. (2023).

**Evaluation**  Following fine-tuning, all models are tested on their respective test datasets, namely CS, CRS, CRSB, GS, GRS, and GRSB. Additionally, the baseline model is evaluated against spatial captioning datasets. For referring expression datasets, we evaluate on average IOU. For pure spatial captioning, we compute caption likelihoods over correct and incorrect captions and consider a selection correct when the model assigns it the highest likelihood. We use this method in an attempt to emulate the evaluation methods in Kamath et al. (2023).

Additionally, we evaluate on the datasets provided by Kamath et al. (2023), namely COO, CTO, GOO, GTO, W1, and W2. For COCO and GQA datasets, we limit our evaluation by grouping shared models and datasets. Models fine-tuned on CS, CRS, and CRSB, for example, are *all* evaluated on each of COO and CTO, while models fine-tuned on GS, GRS, and GRSB are *all* evaluated on each of GOO and GTO. This evaluation is used isolate the effect which fine-tuning on increasingly complex tasks has on simple spatial reasoning tasks across the same image space. For W1 and W2, however, we use all fine-tuned models for evaluation. This evaluation helps us measure and understand the same effects on out-of-distribution image spaces. Lastly, for each of the Kamath et al. (2023) datasets we evaluate the untrained model, where we use the same spatial reasoning metrics to establish its baseline spatial competency against our specialized models. The results of our fine-tuning experiments is available in Table 2.

**Token Analysis** For each dataset, compression metrics 3.1 and SVD analyses 3.2 are extracted from baseline and finetuned models in a manner identical to evaluation. For the COCO and GQA-derived (Kamath et al., 2023) datasets, we group shared models and datasets for token analysis, and for W1 and W2, we conduct token analysis on each dataset using every fine-tuned model. We also conduct token analysis for the baseline model on each of the datasets.

## A.6 LLM USAGE

While LLMs did not contribute to our paper to the extent that we would consider it an author, we did utilize it in creating this work. The primary way we used LLMs, primarily Claude, was in writing code for running our experiments and generating the plots that are present in the paper. However, we note that the ideas themselves and the experiments that we considered were all human driven. For instance consider the synthetic data generation script. We used Claude to write this script, carefully laying out exactly what we wanted generated and how we wanted it done. In writing the paper, we used LLMs for generating some of the latex for the equations and for proof-reading. This proof-reading consisted of identifying any grammatical errors, any breaks of the anonymity guidelines, and recommended areas of the paper that could be improved.

| Dataset | Shorthand | Annotations | | Prepositions | Objects |
|---------|-----------|-------------|---|--------------|---------|
| | | Train | Test | | |
| COCO Spatial | CS | 69,348 | 2,862 | left, right, top, bottom | 1, 2 |
| COCO Ref. Spatial | CRS | 69,348 | 2,862 | left, right, top, bottom | 1, 2 |
| COCO Ref. Spatial Box | CRSB | 69,348 | 2,862 | left, right, top, bottom | 1, 2 |
| GQA Spatial | GS | 66,356 | 9,265 | left, right | 2 |
| GQA Ref. Spatial | GRS | 66,356 | 9,265 | left, right | 2 |
| GQA Ref. Spatial Box | GRSB | 66,356 | 9,265 | left, right | 2 |
| COCO One-Obj | COO | – | 2,247 | left, right, top, bottom | 1 |
| COCO Two-Obj | CTO | – | 440 | left, right, above, below | 2 |
| GQA One-Obj | GOO | – | 1,160 | left, right, top, bottom | 1 |
| GQA Two-Obj | GTO | – | 291 | left, right, front, behind | 2 |
| Whatsup-1 (Cont. Clevr) | W1 | – | 408 | left, right, front, behind | 2 |
| Whatsup-2 (Cont. Images) | W2 | – | 412 | left, right, on, under | 2 |

Table 1: Dataset Overview

Table 2: Fine-tuned Model Performance Across All Datasets

| Dataset | Model | Token Prob. | IOU |
|---------|-------|-------------|-----|
| CS | CS-FT | 0.97 | – |
| | Baseline | 0.95 | – |
| CRS | CRS-FT | – | 0.80 |
| CRSB | CRSB-FT | – | 0.79 |
| GS | GS-FT | 0.99 | – |
| GRS | GRS-FT | – | 0.68 |
| GRSB | GRSB-FT | – | 0.71 |
| COO | CS-FT | 0.99 | – |
| | CRS-FT | 0.88 | – |
| | CRSB-FT | 0.84 | – |
| CTO | CS-FT | 0.99 | – |
| | CRS-FT | 0.90 | – |
| | CRSB-FT | 0.89 | – |
| | Baseline | 0.86 | – |
| GOO | GS-FT | 0.94 | – |
| | GRS-FT | 0.93 | – |
| | GRSB-FT | 0.95 | – |
| | Baseline | 0.97 | – |
| GTO | GS-FT | 0.97 | – |
| | GRS-FT | 0.87 | – |
| | GRSB-FT | 0.89 | – |
| | Baseline | 0.89 | – |
| W1 | CS-FT | 0.49 | – |
| | CRS-FT | 0.69 | – |
| | CRSB-FT | 0.74 | – |
| | GS-FT | 0.50 | – |
| | GRS-FT | 0.52 | – |
| | GRSB-FT | 0.52 | – |
| | Baseline | 0.55 | – |
| W2 | CS-FT | 0.50 | – |
| | CRS-FT | 0.51 | – |
| | CRSB-FT | 0.50 | – |
| | GS-FT | 0.50 | – |
| | GRS-FT | 0.49 | – |
| | GRSB-FT | 0.50 | – |
| | Baseline | 0.50 | – |

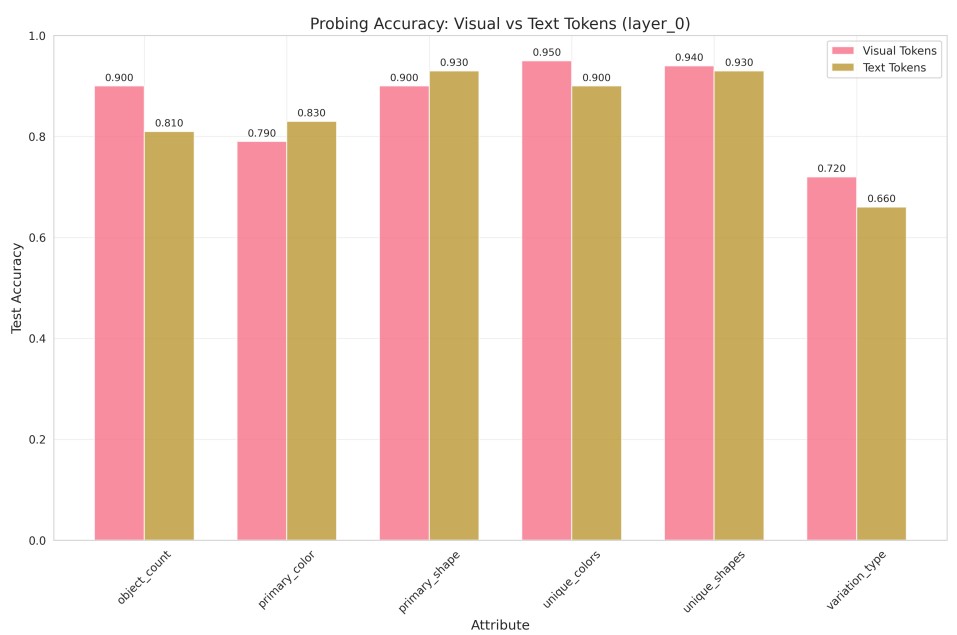

Figure 10: Llama probes

