# OpenReview forum: "Seeing Beyond Redundancy: Task Complexity's Role in Vision Token Specialization in VLLMs"
_ICLR.cc/2026/Conference — Submitted to ICLR 2026_

### Official Review · Reviewer_SLXB · 2025-10-28

**Soundness:** 2
**Presentation:** 3
**Contribution:** 2
**Rating:** 4
**Confidence:** 3

**Summary:**

This paper investigates the visual redundancy problem of large vision language models through controlled experiments. Specifically, it constructs a synthetic benchmark with a set of metrics to measure and understand the problem. Fine-tuning is also employed to investigate the impact of data. Results reveal the connection between task complexity and visual compression.

**Strengths:**

1. The paper presents a comprehensive suite of metrics accompanied by extensive empirical analyses to characterize visual information compression across layers, providing valuable insights

2. Detailed ablation studies are provided

3. Fine-tuning is included for a deeper understanding

**Weaknesses:**

1. The paper lacks sufficient justification for the choice of evaluation metrics. A more thorough discussion of the theoretical foundations and practical motivations underlying these metrics would strengthen the methodological framework.

2. The experimental validation is limited to two models (Molmo-7B and Llama 3.2). The generalizability of the findings could be substantially improved by including models from varying parameter scales.

3. The zero-shot analysis relies exclusively on synthetic datasets with simplified characteristics. The extent to which these findings translate to real-world scenarios remains insufficiently addressed.

4. The practical impact of this work would be considerably enhanced if the derived insights were used to fine-tune vision-language models with improved compression efficiency and reduced redundancy.

**Questions:**

1. The terminology "Large Vision-Language Model (LVLM)" appears to be more prevalent in the literature than "Vision Large Language Model (VLLM)."

2. The citation format requires attention to stylistic conventions. Several in-text citations currently employ \cite without parentheses, whereas \citep would be more appropriate. Instances include lines 104-105 where the citations serve as supplementary support rather than grammatical subjects.

3. Typo in line 311 "Figure 2 provides further insights into have visual compression is correlated"

---

> ### Author Response · Authors · 2025-11-29
>
> We appreciate your review!
>
> Alternative Models: Please see the discussion in the general comment above.
>
> Real-world Zero-shot Results: Please see our updated COCO results in the general comment above.
>
> Applying our work to fine-tuning/compression: Please see the discussion of leveraging our metrics for compression in the general comment.
>
> Terminology: We have noted in the main text this potential alternative term for the models that we used in the paper.
> Citation Format: Thank you for bringing this to our attention; we have updated the citation format.

---

### Official Review · Reviewer_HqRP · 2025-10-31

**Soundness:** 3
**Presentation:** 2
**Contribution:** 2
**Rating:** 4
**Confidence:** 4

**Summary:**

This paper investigates visual redundancy and compression phenomena in VLMs, with a focus on the relationship between task complexity and vision token specialization. The authors construct a synthetic dataset to systematically vary visual complexity and propose novel compression/ redundancy measurement metrics, including norm-based and rank-based measures, SVD alignment analyses, linear probe evaluations, and token ablation experiments. They conduct zero-shot and fine-tuning experiments on Molmo and LLaMA-v3.2-Vision, analyzing how task type (referring expression vs spatial reasoning) and dataset complexity influence internal representations and compression behavior. The paper concludes with proposed compression strategies and implications for fine-tuning.

**Strengths:**

1. Systematic metric design – The work proposes a comprehensive suite of metrics (both norm- and rank-based, plus SVD alignment) to analyze compression and redundancy in VLLMs’ hidden states, offering more granular insight than prior attention-based analyses.

2. Detailed layer-wise analysis – The visualization across layers for different metrics provides an interpretable picture of how visual information is redistributed within models.

3. Task complexity perspective – The link between downstream task complexity and optimal compression levels is well articulated and supported by multiple evaluation angles.

**Weaknesses:**

1. Synthetic dataset reliance – The main analyses are conducted on a fully synthetic dataset designed by the authors, with limited validation of whether the findings generalize to real-world tasks. The COCO and GQA datasets used in the synthetic data experiments were also only analyzed using the metrics proposed in the paper, rather than through more intuitive computations of prediction accuracy.

2. Evaluation metric coverage vs accuracy gains – The paper heavily focuses on reporting compression/ redundancy metrics but lacks direct evidence that these methods can improve benchmark accuracy when applied in compression policies. A simple empirical demonstration of accuracy improvement would make the contribution more tangible.

3. Architectural limitation in scope – Both Molmo and LLaMA-3.2-Vision adopt CLIP-style fixed-resolution vision encoders with image patching (slice into fixed-size tokens). Newer architectures (e.g., Qwen-VL, GLM-VL) use native resolution and dynamic tokenization according to input resolution, potentially altering redundancy/compression behavior. The generality of the conclusions under these architectures is not assessed.

4. Overlap with prior work’s findings – Some behavioral observations[1] have been highlighted in several earlier VLLM diagnostic studies. The novelty claim would benefit from a clearer positioning relative to these works.

[1] Label Words are Anchors: An Information Flow Perspective for Understanding In-Context Learning

**Questions:**

See weaknesses.

---

> ### Author Response · Authors · 2025-11-29
>
> Thank you for your feedback!
>
> Synthetic dataset reliance: Please see the general comment for updated results on COCO.
>
> Evaluation metric coverage vs accuracy gains: The goal of our work was not to boost the accuracy of compression methods, although that might be a potential application for these metrics. Rather, we wanted to point out the potential problems with the way that these models process visual information. A lot of prior work focuses on increasing compression, but we believe that rather than taking advantage of the compressibility, we should focus on obtaining models that compress information less, leading to better performance on more complex tasks.
>
> Architectural limitation in scope: Please see our discussion in the general comment for more justification of our architectural selection.
>
> Overlap with prior work’s findings: There are indeed some works in the LLM space that uncover potentially related findings. However, VLLMs (or LVLMs) have not been studied to the same extent, and we believe adding visual information is a substantial enough modification to these models that it is not clear whether works that exclusively focus on LLMs would generalize.

---

### Official Review · Reviewer_pQNZ · 2025-10-31

**Soundness:** 3
**Presentation:** 3
**Contribution:** 2
**Rating:** 4
**Confidence:** 2

**Summary:**

The aim of this work is to propose an analysis on why MLLM still struggle with visually fine grained tasks even if they excel at tasks involving the global image semantic. The authors explore this via statistical analysis on the visual and text tokens in intermediate layers of the LLM as well as different probing mechanisms like training FC classifiers on top of the tokens and randomly dropping them to see impact on performance. Most experiments are run on Molmo on a synthetic dataset created ad hoc from the authors. Few fine tuning experiments use real data. The findings are that there is a high redundancy within the visual tokens on a LLM and that they tend to be optimized for general vision tasks and not fine grained ones. Moreover, If the model are fine tuning on challenging localization tasks most of the representation changes are in the text part of the model and not in the multimodal one.

**Strengths:**

+ Authors propose a very detailed statistical analysis to uncover redundancy in the tokens representations within a LLM. A lot of the technique proposed could likelly be re-used for other works interested in uncovering more about the hidden representation of these models.

+ Surprising finding that in the experimental settings of the work fine tuning a model on visual data seems to  overwhelmingly alter text representations while leaving vision representations largely unaltered.

+ Carefully curated creation of synthetic data to support the experimental analysis in the paper.

**Weaknesses:**

a. **Limited experimental analysis**: Most experiments of the paper are performed using only the Molmo MLLM, would have been interesting to see the analysis expanded to other models trained on different data mixtures and with different architectures. The paper does consider llama, but only for the experiments on probes and visual ablations. The analysis of other decoder based MLLM besides Olmo would have made this submission more strong.

B. **Nice analysis, but limited applicability**: While the work does provide some nice insights the findings are not very actionable and mostly provide experimental evidence of behavior that is quite known to practitioners. Namely: that the amount of tokens that can be dropped from a LLM input is a function of how “hard” a task is and that an effective post-training strategy can involve only the LLM without touching the visual encoder in the model. Also the observation that harder visual task will bring more changes in the model can likelly be linked to the perplexity for the model on those tasks while training. If on average the tasks are harder for the model they could cause higher gradients which in turn results in a bigger shift in the text components of the visual model.

C. **Small scale controlled experiments**: The paper does a nice job at creating a setting where the claim can be tested and isolated, but it does not verify whether the claim holds on bigger and more realistic settings. For example the token analysis in Sec. 4.1 is all performed only on synthetic images, while the fine tuning experiments in Sec. 4.2 consider fine tuning only on (few) visual tasks, while in practice most MLLM would be fine tuned on a way bigger mixture of visual and textual tasks. Exploring what happen in the more realistic settings would have made the submission stronger.

**Questions:**

1. What’s the text prompt for the synthetic dataset you generated?

Few typos
L311: “into have”
L481: “are more require”
Across the paper you read few times “muiltimodal” instead of “multimodal”

---

> ### Author Response · Authors · 2025-11-29
>
> We appreciate your feedback!
>
> Limited experimental analysis: Please see a discussion of our model selection in the general comment above.
>
> Nice analysis, but limited applicability: Please see the discussion of leveraging our metrics for compression in the general comment.
>
> Small scale-controlled experiments: Please see our section in the general comment above on our new COCO results.
>
> Text prompt for creating synthetic data: Our synthetic data was generated using PIL in python. We clarified this in the paper. If you are asking about the prompts that were used for our analysis of this dataset, we also updated the paper to include this in the Appendix along with a list for COCO as well.

---

### Official Review · Reviewer_QipL · 2025-11-02

**Soundness:** 3
**Presentation:** 3
**Contribution:** 3
**Rating:** 4
**Confidence:** 2

**Summary:**

This paper conducts an in-depth study on the problem of visual information redundancy in vision-language large models (VLLMs), and points out that visual information redundancy is one of the important reasons for the poor performance of the model in complex visual tasks, such as fine-grained object recognition and spatial reasoning. The author constructed a synthetic dataset, quantified the complexity of tasks, and found that complex tasks (such as object counting) require more specialized visual tokens, have lower redundancy, and are sensitive to compression. However, simple tasks (such as color recognition) are not sensitive to redundancy and can even tolerate up to 99% token discard. Through fine-tuning experiments on the model, the author found that fine-tuning mainly altered the text representation of the model, while the visual representation changed relatively little. Moreover, different types of tasks (spatial reasoning vs. object localization) affected the internal representation of the model in different ways. Based on these findings, the authors proposed compression strategies and training suggestions for VLLMs, namely, appropriately compressing information in the early layers, carefully compressing in the middle layers, reducing the compression ratio for complex tasks, significantly compressing for simple tasks, and paying more attention to the updates of text and multimodal projection layers during fine-tuning.

**Strengths:**

- This paper proposes and comprehensively applies multiple quantitative indicators (such as Gini coefficient, stable rank, participation rate, etc.) to systematically analyze visual information redundancy from the two levels of token norm and matrix rank, surpassing previous studies that only focused on attention distribution and providing a more comprehensive tool for understanding the internal visual information processing of VLLMs.

- The experiments precisely controls variables through the construction of synthetic datasets, the negative correlation between task complexity (such as the number of objects and the difficulty of spatial reasoning) and the degree of visual information redundancy was clearly verified for the first time, providing direct evidence for explaining the performance bottleneck of VLLMs in complex visual tasks

**Weaknesses:**

- Some findings, such as "there is a connection between task complexity and visual compression", are similar with the conclusions given in previous works like PDrop[1].

- Fine-tuning experiments are only based on simplified subsets of COCO and GQA (such as objects with only "left-right" relationships), and more complex spatial relationships (such as spatial reasoning in ERQA) have not been tested, which may underestimate the model's redundant performance in real complex tasks.

- The experiment mainly uses syntheti5c data of simple geometric shapes (fixed color/shape/size), lacking complex factors such as texture, occlusion, and lighting changes in real images, which may lead to insufficient generalization of the conclusion in real scenes.

[1] Xing, et al. Pyramiddrop: Accelerating your large vision-language models via pyramid visual redundancy reduction. CVPR, 202

**Questions:**

- How to specifically configure the vision token compression on the task with different complexities? And, will the configuration setting be quite different among different types of models?

---

> ### Author Response · Authors · 2025-11-29
>
> Thank you for your review!
>
> Comparison to PyramidDrop: PyramindDrop is one, among a number of works, that has at some level reached a similar conclusion to our own and inspired the experiments that we present in our paper. However, most of these works, PyramidDrop included, focus more on the development of a compression technique and less on understanding the nuances of how information is compressed in certain images. For instance, they make a similar observation at a task level that more complex tasks are less compressible and therefore have less token redundancy, but we are focused more on the individual image level to understand which image features specifically lead to more redundancy, providing a greater level of detail and a more targeted analysis than prior works.
>
> Results on Real Data: Please see the general comment for a discussion of additional experiments targeting COCO.
>
> Question on how to configure token compression: Please see the discussion of leveraging our metrics for compression in the general comment.

---

### Author Response · Authors · 2025-11-29
**General Comment**

We would like to thank all of the reviewers for their helpful feedback and comments.

The most common ask was to expand some of our analyses to more complex real-world datasets. Included in the updated version of the paper are zero-shot COCO results. Many of the general trends follow those observed in the synthetic dataset with some interesting new insights. A detailed discussion and comparison can be found throughout the zero-shot experimental sections along with the visual ablations in the Appendix due to space constraints. We believe this has substantially improved our work.

Another common ask was to expand our analyses to other models. Qwen was indeed a target for us prior to submission; however, it is fairly time intensive to apply the techniques to new models as specific alterations are required to accommodate each model’s pre-processing strategy and architecture. However, we will note that many of the prior visual compression works that have loosely drawn similar conclusions and inspired our work were done with LLaVA and we specifically tried to target more recent models that were most dissimilar to one another, where Molmo is more similar to Qwen/LLaVA and LLaMA is distinct in that it leverages cross-attention.

The last general feedback that occurred across multiple reviews was that we should leverage some of our insights and metrics for improving visual compression. We did not opt to do this so that we could preserve most of the space in the paper for a robust analysis of compression in VLLMs. Indeed, we felt that prior analyses would oftentimes only be performed only enough to motivate a new compression method, leaving more detailed insights untouched. For instance, rather than just saying that complex tasks can be compressed less, our work provides specific insights into a number of different tasks and how each one scales as tokens are removed from the model. Therefore, the motivation of our work was to target these scenarios. While outside of the scope of our current work, a natural extension of it would be to leverage some of the metrics that we propose to select the best compression regime. This would involve operating at a dataset/task level, averaging results over samples to determine which layers of the model to compress and how much to compress them. If multiple tasks are being considered, different settings for various tasks could be selected to ensure maximum compression.

---

### Meta-Review · Area_Chair_wAb8 · 2025-12-22

**Summary:**

This paper investigates the phenomenon of visual redundancy in Vision Large Language Models (VLLMs), focusing on the relationship between task complexity and vision token specialization. The authors construct a synthetic benchmark of geometric shapes to probe visual features and introduce a suite of metrics (including norm-based, rank-based, and participation rate measures) to quantify redundancy. Through zero-shot and fine-tuning experiments on models like Molmo and Llama-3.2-Vision, the study concludes that complex tasks (e.g., spatial reasoning, object counting) exhibit lower redundancy and higher sensitivity to token pruning compared to simple tasks. Additionally, the paper observes that fine-tuning primarily alters textual representations rather than visual ones.

Reviewers acknowledged the systematic metric design and the intuitive verification that task complexity correlates with token specialization. However, the consensus among reviewers was that the paper suffers from limited practical applicability and an over-reliance on synthetic data. Critically, reviewers felt the work remained observational without demonstrating how these insights could be leveraged to actually improve model performance or compression efficiency. Furthermore, the restriction to fixed-resolution architectures (excluding modern dynamic-resolution models like Qwen-VL) limited the generalizability of the findings. Despite the inclusion of zero-shot COCO results in the rebuttal, the core concerns regarding the scope and impact of the contribution remained unresolved. Therefore, the Area Chair (AC) recommends rejection.

**Reviewer Concerns:**

Several key weaknesses that were not fully addressed during the rebuttal:
- A primary critique shared by Reviewers pQNZ, HqRP, and SLXB was that the paper stops at analysis. While the metrics are interesting, the work does not propose a concrete method to utilize these insights for improving accuracy or efficiency. Reviewers argued that the contribution would be significantly stronger if the authors demonstrated that their metrics could guide a better compression policy or fine-tuning strategy, rather than just observing that "harder tasks need more tokens". The authors' rebuttal decision to *not* include such an application to "preserve space for analysis" was unconvincing to reviewers looking for tangible impact.
- Reviewers QipL, pQNZ, and SLXB expressed concern that the findings are heavily derived from a synthetic dataset of simple geometric shapes. They questioned whether conclusions drawn from such simplified data (lacking texture, occlusion, and lighting variations) generalize to real-world visual complexities.
- Reviewers HqRP and pQNZ noted that the analysis is restricted to models with fixed-resolution patching (Molmo, Llama-3.2). They argued that newer architectures like Qwen-VL or GLM-VL, which use dynamic resolution and tokenization, might exhibit fundamentally different redundancy patterns. The authors' rebuttal that adding these models was "time intensive" did not alleviate the concern that the findings might be architecture-specific.

**Reviewer Scores:**

- Reviewers viewed the work as a diagnostic study that confirms existing intuitions (e.g., "complex tasks allow less compression") rather than a transformative piece of research. The lack of a proposed algorithm or a surprising counter-intuitive finding limited the perceived novelty.
- The combination of synthetic data and a limited set of architectures left reviewers unconvinced that the "rules of redundancy" proposed here would hold up in the broader, rapidly evolving landscape of VLLMs.

---

### Decision · Program_Chairs · 2026-01-26

Reject